# First Report of Antimicrobial Susceptibility and Virulence Gene Characterization Associated with *Staphylococcus aureus* Carriage in Healthy Camels from Tunisia

**DOI:** 10.3390/ani11092754

**Published:** 2021-09-21

**Authors:** Faten Ben Chehida, Haythem Gharsa, Wafa Tombari, Rachid Selmi, Sana Khaldi, Monia Daaloul, Karim Ben Slama, Lilia Messadi

**Affiliations:** 1Laboratory of Microbiology, Immunology and General Pathology, Institution of Agricultural Research and Higher Education, National School of Veterinary Medicine of Sidi Thabet, University of Manouba, Sidi Thabet 2020, Tunisia; wafatombari@yahoo.fr (W.T.); selmiveto1983@gmail.com (R.S.); moniajd2017@gmail.com (M.D.); lilia_messadi@yahoo.fr (L.M.); 2Laboratory of Microorganisms and Active Biomolecules, Higher Institute of Applied Biological Sciences of Tunis, University of Tunis El Manar, Tunis 2092, Tunisia; haythemgharsa@yahoo.fr (H.G.); karim.benslama@issbat.utm.tn (K.B.S.); 3Veterinary Service, General Directorate of Military Health, Ministry of National Defense, Tunis 1008, Tunisia; 4Department of Sciences and Pathology of Animal Reproduction, Institution of Agricultural Research and Higher Education, National School of Veterinary Medicine of Sidi Thabet, University of Manouba, Sidi Thabet 2020, Tunisia; sana.khaldi@gmail.com

**Keywords:** *Camelus dromedarius*, Tunisia, *Staphylococcus aureus*, virulence, typing, antimicrobial resistance

## Abstract

**Simple Summary:**

The one-humped camel (*Camelus dromedarius*) is an important livestock species and is present in more than 46 national entities, with 80% of the camel population inhabiting Africa. In these regions, the role of camels in the livestock economy is highly valuable and a part of this camel herd is valorized on national or international markets for meat production, live animal export or milk production. Even if camels are the species that is most adapted to the harsh conditions of arid/semi-arid rangelands, they can be susceptible to a high number of pathogens, including *S. aureus*. This latter is often associated with asymptomatic carriage but can also be responsible for several diseases, therefore causing considerable economical losses. Continued monitoring and control assume particular importance in containing the spread of the bacterium since it constitutes an important zoonotic disease.

**Abstract:**

A total of 318 nasal and rectal swabs were collected from 159 apparently healthy camels (*Camelus dromedarius*) randomly selected from five regions in southern and central Tunisia and screened for *Staphylococcus aureus* carriage. *Staphylococcus* spp. were recovered from 152 of 159 camels studied (95.6%) and in total 258 swabs (81%) were positive. Among these isolates, 16 were coagulase positive *Staphylococcus* (CoPS) (6.2%) and were characterized by biochemical and molecular tests as *S. aureus*. These were isolated from 14 camels (8.8%) with co-carriage in nasal and rectal mucosa by two camels. All *S. aureus* isolates recovered were methicillin-susceptible Staphylococcus aureus (MSSA) and were characterized by *spa* typing and PFGE. Three different *spa* types were recovered: t729, t4013 and a *spa* type newly registered as t19687, which was the most common. PFGE analysis revealed seven different patterns and these were characterized by MLST, which revealed five different sequence types (ST6, ST88, ST3583 and two new sequences, ST6504 and ST6506). All isolates harbored different virulence genes, including *hld*, encoding delta hemolysin; *luk*E–*luk*D, encoding bicomponent leukotoxin LukE–LukD; the *clf*B gene, encoding clumping factor B; the *laminin* gene, encoding laminin-binding protein; and *cap*8, encoding capsule type 8. Fifteen isolates harbored hemolysin beta (*hlb*) and fourteen encoded hemolysin alpha (*hla*) and hemolysin G2 (*hlg_v_*). Adhesin factors, including *clfA* and *fnbB*, were detected in five and four isolates respectively. Binding proteins, including collagen (*cbp*) and elastin-binding protein (*ebp*), were detected in two *S. aureus* isolates while fibrinogen-binding protein (*fib*) was identified in four isolates. This study provides the first set of genotyping data on the population structure and presence of toxin genes of *S. aureus* strains in Tunisian camels.

## 1. Introduction

According to the most recent data available in 2020, the global population of dromedary camels (*Camelus dromedarius*, one-humped camel) is around 35 million, with a distribution limited to the African and Middle Eastern countries [1]. Cameline livestock in Tunisia, the number of which is estimated to comprise 80,000 female units, are mainly located in the southern and central parts of the country [2]. Tunisian camels are commonly kept under traditional pastoral production systems and they make the best use of this production system thanks to their morphological and physiological particularities. Camels are indeed able to adapt not only to meteorological constraints (aridification of the environment), but also to the zootechnical changes brought by more intensive farming systems emerging from increasing urbanization [3]. The ability of camels to survive in arid and semiarid areas, with high potential to convert the scant resources of the desert into milk and meat, makes them a valuable livestock species, providing a significant percentage of the population with animal protein and milk [4].

Camels were formerly thought not to be affected by most of the diseases commonly impacting livestock. However, recent data have confirmed their susceptibility to a high number of pathogens [5]. Moreover, the increased consumption of and contact with camel meat and milk represent a serious source for zoonotic disease transmission to humans. 

Staphylococci in general, and *S. aureus* in particular, are listed among the most important camel zoonotic diseases and infections reported worldwide [6]. *S. aureus* is a common and widespread bacterium usually associated with asymptomatic carriage, which colonizes the skin, nares and other mucosa of various animal species [7]. However, when a decrease in immunitary efficiency is noticed, it may behave as an opportunistic pathogen, causing a wide variety of diseases ranging from skin and soft tissue infections to sepsis and toxic shock [8]. 

In ruminants and pseudoruminants, such as dromedaries, *S. aureus* is usually associated with mastitis, skin infections (including skin abscesses and necrosis), respiratory tract diseases (pneumonia) and endometritis; it therefore impacts the livestock productivity, resulting in considerable economical losses [9,10,11,12,13].

Isolation of the bacterium from both healthy and diseased camels has been reported by a number of researchers worldwide. However, the lack of information about the carriage of *Staphylococcus* spp. in dromedaries in Tunisia launched the present investigation of this animal species.

The objectives of this study were to determine the nasal and rectal carriage of *Staphylococcus* spp., especially *S. aureus*, in healthy dromedaries in Tunisia, to carry out the molecular typing of the recovered isolates and to determine their antimicrobial resistance profile and virulence genes.

## 2. Results

### 2.1. Global Carriage of Staphylococcus spp. and S. aureus in Camels

*Staphylococcus* spp. isolates were recovered from 152 (95.6%) of the 159 apparently healthy camels studied. A total of 258 swabs from the 318 swabs collected (81%) were positive for *Staphylococcus* spp., among which 139 (53%) were obtained from nasal cavities and 119 (46%) from rectal samples. Of these isolates, 242 were found to be coagulase-negative (93.7%), while only 16 were coagulase-positive (6.2%). 

All CoPS isolates were subject to further testing for formal species identification, using both conventional and species-specific PCR methods. All 16 isolates were confirmed as belonging to the species *S. aureus* and only 4 among them (25%) were obtained from rectal samples, while the remaining isolates (12) had a nasal origin (75%). From the total population of camels (159), *S. aureus* was detected in 14 camels (8.8%), since co-carriage of *S. aureus* in the nasal and rectal mucosa was observed in two animals.

### 2.2. Resistance Genes and Virulence Markers among S. aureus Isolates

All *S. aureus* isolates were phenotypically susceptible to the tested antimicrobial agents. After further investigation, the *mec*A gene defining MRSA was not detected, but we could discern the common virulence markers (Table 1), since all MSSA isolates carried the *hld* gene encoding delta hemolysin, the *luk*E–*luk*D gene encoding the bicomponent leukotoxin LukE–LukD, the *clf*B gene encoding the clumping factor B, the *laminin* gene encoding the laminin-binding protein and *cap*8 encoding the type 8 capsule. The other virulence genes carried by MSSA isolates were *hlb* (15 isolates, 93.7%), *hla/hlg_v_* (14 isolates, 87.5%), *ebp* (10 isolates, 62%), *clf*A (5 isolates, 31%), *fnb*B/*fib* (4 isolates, 25%) and *cbp* (2 isolates, 12.5%). On the other hand, no genes encoding for the staphylococcal enterotoxins were retrieved in any of the *S. aureus* isolates, nor were the nine virulence genes *siet, eta, etb, fnbA, PVL S* and *F, lukM, hlg, bsp* and *cap*5.

### 2.3. Characteristics of MSSA Detected in This Study

The results of bacterial typing of the 16 MSSA isolates individualized in this study are detailed in Figure 1. All of these isolates were submitted to spa typing, which allowed the detection of three different *spa* types: t729, t4013 and a newly registered *spa* type, t19687, the last being the most common. The analysis of their *Sma*I macro-restriction profiles revealed seven different PFGE patterns. MLST was conducted for seven representative MSSA (per PFGE profile) isolates showing five different STs: ST6, ST88, ST3583 and two STs newly registered as ST6504 and ST6506. Table 2 summarizes the characteristics of the 16 *S. aureus* isolates recovered from 14 healthy camels in this study. 

## 3. Discussion

Although *S. aureus* is a well-known bacterial pathogen incriminated in human and animal infections, little information is currently available about its occurrence, antibiotic resistance and toxinogenic potential in dromedaries.

We thus undertook the present study in order to investigate the presence of *Staphylococcus* spp. in camels’ nostrils and rectum, as well as to assess the prevalence of *S. aureus* among the confirmed isolates, which allowed an initial insight into population characterization, and the relative abundances of virulence-associated genes of *S. aureus* in camels.

To our knowledge, this is the first study regarding the carriage of staphylococci in camels performed in Tunisia. Previously, we conducted studies in other species, including farm animals and pets [14], sheep [15], goats [16] and donkeys [17]. In our study, a relatively high prevalence of carriage of *Staphylococcus* spp. in the nasal and rectal mucosa (95.6%) was observed in healthy camels, particularly in nasal swabs. This observation can be explained by the fact that commensal staphylococci are mainly found in the upper respiratory tract of animals, and they play a pathogenic role when the general resistance of the host decreases [18].

However, lower respiratory carriage rates of *Staphylococcus* spp. in healthy camels have been reported in previous studies carried out in Africa and the Middle East, including Kenya, Iran, Sudan and Ethiopia (34%, 29%, 30.4% and 12%, respectively) [19,20,21]. In Sudan and Ethiopia, camels with pneumonic lesions were also positive for *Staphylococcus* spp. (30.4% and 26.5%, respectively) [18,21]. Very few studies have focused on the carriage of *Staphylococcus* spp. in rectal mucosa; Al-Thani et al. (2012) could not report any isolates in the rectums of healthy camels [22].

Coagulase production by *Staphylococcus* species is considered a virulence factor that enables the pathogen to evade the host’s immune system and it is frequently used to identify *S. aureus* [23]. Indeed, compared to coagulase-negative staphylococci, the coagulase-positive species group is commonly associated with severe infections [24]. In our study, a relatively low recovery rate of CoPS was noticed in the nasal and rectal samples collected from healthy camels, and all were identified as *S. aureus*. This latter is acknowledged to be a commensal colonizer of the skin, nose and mucous membranes of healthy humans and animals, but also to be an opportunistic pathogen in several infectious diseases [25]. In fact, *S. aureus* is the most important CoPS species due to its combination of toxin-mediated virulence, its invasiveness and its antibiotic resistance, and it is a major causative agent of several pathologies in animals [26]. It is mostly associated with mastitis, suppurative dermatitis, abscesses, arthritis, endometritis and respiratory infections in camels [27].

In our study, *S. aureus* was more commonly retrieved in intensive farming units located in Hamam Sousse and Bouficha, which are touristic areas. This observation can be explained by the intensification of breeding in the region of Sahel for purely touristic purposes and the closer proximity of individuals that this generates, especially when associated with poor hygiene conditions. In fact, in these farms, the high density of animals and their use in densely populated urban areas favor the colonization of nasal cavities by coagulase-positive staphylococci, with possible zoonotic transmission, particularly when close contact between humans and camels occurs, as it can do on various occasions during watering, riding, grooming and milking [28]. Human-to-animal transmission of *Staphylococcus* can been suggested, since human-related clonal lineages have been identified in *Staphylococcus* strains from non-human primates, goats, sheep, poultry and pets [25]. 

The isolation rate of *S. aureus* in dromedaries in our study was relatively low (6.2%) compared to the other species in Tunisia, such as in domestic carnivores (6.5%), goats (16.5%), sheep and ewes (44.8% and 26.9%) and donkeys (50%) [14,15,16,29,30]. Moreover, *S. aureus* was predominantly recovered from the nasal samples (75%) rather than the rectal samples (25%). These different carriage rates could be at least partially explained by self-care behavior, such as nose licking, and/or by different kinds of farm management. In humans, anterior nares are the main ecological niche for *S. aureus*, which persistently or intermittently colonizes the nares of 30 to 50% of healthy adults [31]. The study of the human nasal microbiota has shown that there is competition between *S. aureus* and other bacteria, like coagulase-negative staphylococci, through the production of antimicrobial molecules, and the same phenomenon could probably explain the relatively low rate of *S. aureus* carriage in camels, given the high carriage rate of coagulase-negative staphylococci [32].

Currently, the scientific literature shows that the prevalence of *S. aureus* nasal carriage in camels varies between countries. Previous work carried out on healthy camels in some African and Middle Eastern countries showed higher nasal carriage of *S. aureus*, including in Algeria (53%), Jordan (13.7%), Qatar (43.9%), Saudi Arabia (56.3% and 89.1%) and Nigeria (14%) [6,22,33,34,35,36,37]. A prevalence of 19.2% was registered in Sudan, the samples being nasal swabs collected from camels with clinical signs of pneumonia [38]. The range of the carriage rates reported in these different countries is large, and these discrepancies may partially originate from the differences in the sampling quality and in the bacterial culture techniques used in these studies.

All the currently available data concur that *S. aureus* resides as a normal inhabitant of the upper respiratory tract, and several studies show that the nares are the most consistent area from which this organism can be isolated. The carriage of *S. aureus* in the nasal cavities appears to play a key role in the epidemiology and pathogenesis of pulmonary infection, very likely causing secondary pneumonia in immunocompromised animals [21,39].

Over the past decade, the problem of antimicrobial resistance in the African continent has attracted a special interest [40]; however, little is known about the real extent of the problem, especially in camels, since few studies have focused on this species. In the present study, resistance genes were absent in *S. aureus* isolates, which revealed their susceptibility to all the tested antimicrobials. Furthermore, MRSA was not detected among the tested isolates, no were any conspicuous resistance markers, such as *van* genes (glycopeptide resistance). In fact, it was demonstrated that MRSA isolates generally have resistance to other non-beta-lactam agents in addition to methicillin resistance, while MSSA isolates show susceptibility to most tested antimicrobials [25]. This high susceptibility among *S. aureus* isolates recovered from camels is remarkable, and it is likely explained by the fact that camels are sturdy animals and are therefore exceptionally treated. This finding is in contrast with the high frequencies of resistance reported in other species.

*Staphylococcus aureus* produces a wide variety of exoproteins that contribute to its ability to colonize and cause disease in mammalian hosts. The pathogenesis of this organism relies on the production of an arsenal of virulence factors [41]. To our knowledge, the prevalence of virulence genes among camels’ isolates has never been investigated in Tunisia, and only scarcely in other countries, which makes it difficult to compare our results with previous data. Isolates’ virulence profiles were very similar, and hemolysin genes, the clumping factor, the capsule gene and leukocidin DE were very common and harbored by all isolates.

The four classes of hemolysins (alpha, beta, delta and gamma) were detected in our isolates; it is known that these toxins are produced by most *S. aureus* strains [42]. hld was identified in all isolates, which was consistent with Dinges et al. (2000) and with studies conducted in Algeria and Saudi Arabia that reported higher than 97% positivity in camel nasal and meat isolates [36,43]. Furthermore, *hlb* has been described as having a human specificity, suggesting a human origin and an adaptation to its animal host [43,44]. The combination *hla–hlab–hld–hlg_v_* was the most prevalent among our strains (75%), and these genes might be either located in the same or associated genetic elements.

Regarding its pathogenicity, *S. aureus* produce up to five different leukocidins: *luk*D–*luk*E, in particular, is one of the only leucocidins to exhibit broad activity in a wide variety of cell types from various species, and it was harbored by 100% of our strains. In fact, it has been reported that this protein possesses a substantial similarity to *lukSF-I* and *lukPQ*, encoded respectively by *S. pseudintermedius*, associated with infections in dogs, and *S. aureus*, associated with infections in horses [45]. In Algeria, Saudi Arabia and the United Arab Emirates, *luk*E–*luk*D was reported in 43.4%, 100% and 55% of camels’ nasal and meat isolates, respectively [9,36,43].

Africa is considered endemic for Panton–Valentine leukocidin (PVL)-positive MSSA isolates, especially in humans. Nevertheless, PVL was not reported in our study: indeed, it is commonly admitted that this protein is produced by 2 to 3% of strains and it is known to be rare in animals [46]. However, Agabou et al. (2017) and Raji et al. (2016) reported the existence of PVL in MSSA and MRSA camel isolates and stated that the presence of this cytotoxic virulence factor in animals must be taken into consideration by public health professionals, given its high pathogenicity in humans and animals [36,43]. The same studies reported the presence of *lukM* in 4.5% of the studied camels’ strains, whereas its absence in our isolates was noticed. This latter has been predominantly found in *S. aureus* incriminated in bovine infections, as a phage-borne *lukPV/lukM* has been proven to encode a bi-component leukotoxin highly active against bovine neutrophiles [45,47]. 

The staphylococcal capsular polysaccharides increase the resistance of bacteria to phagocytosis by polymorphonuclear leucocytes by expressing either the *cap5* or *cap8* genes [48], and all our isolates harbored the *cap8* gene. Similar results were found in Algeria and Saudi Arabia with, respectively, 78.2% and 100% [36,43]. In Algeria, *cap5*, which is predominantly harbored by human isolates and is rare in animal’s strains, has been reported with a low recovery rate (21.7%) [36], but it was not detected in our study. Shuiep et al. (2009) reported that 85% of *S. aureus* isolates from healthy camels’ raw milk in Sudan possessed the *cap5* gene [49].

In addition, *S. aureus* expresses up to 25 different cell wall-anchored (CWA) proteins. ClfA, ClfB and FnbB are among the most important bacterial adhesins and contribute to initiating infection [26]. It has been shown in previous research that ClfB facilitates the colonization of the nasal cavities through its high-affinity interactions with the cornified envelope in the anterior nares, and that this gene is carried by most strains of *S. aureus* [50,51]. Our results are in concordance with those divulged by Agabou et al. (2017), who found that 100% of camel nasal strains carried the *clf*B gene [36]. Our study allowed us to demonstrate that this binding protein may also promote rectal colonization. Others binding proteins were recovered at a low rate, including ClfA, FnbB, Fib and EBP. 

Furthermore, none of the isolates found in our study were positive for the remaining toxin genes tested, including enterotoxins and exfoliative toxins. In other studies, enterotoxins were isolated with low recovery in camels, and only *sec* was recently reported in three *S. aureus* isolates from pasteurized camel milk, while *seg* and *seh* were retrieved in camel nasal samples at frequencies of 4.3% and 17.4%, respectively [36,52]. The occurrence of multiple toxinogenic genes in *S. aureu*s is considered to be rare, which may explain the absence of many of these genes. 

The molecular typing of our *S. aureus* isolates by MLST, PFGE and *spa* typing exhibited little variability among our strains. The analysis of the *Sma*I macrorestriction profiles of the 16 MSSA isolates revealed seven different PFGE patterns that we classified in seven categories from S1 to S7, each defining a clone, and the highest detection rate was registered for the pulsotype S1 (56.3% of isolates, nine isolates). Furthermore, the PFGE analysis showed high clonality among the strains with the same *spa* types. Furthermore, MLST genotyping, widely used to investigate the dynamic nature of *S. aureus* by providing a highly efficient molecular characterization, allowed us to identify five sequence types (ST6, ST88, ST3583 and two newly registered types, ST6504 and ST6506) among the MSSA isolated from camel nasal and rectal swabs. Our work thus described for the first time the presence of these STs in isolates originating from healthy camels. Few studies have focused on the molecular characterization of MSSA isolates in camels, and when they have the presence of other STs was reported, including ST152 and ST1278 in cameline nasal swabs in Algeria [36]; ST1156 in camel meat from Saudi Arabia [43]; ST15, ST1153 and ST130 from Egyptian camels’ milk [53]; and ST1755 in the Emirates [9].

It is useful to note that the ST6 and ST88 spa types described in our study have also been identified in MSSA and MRSA isolates from other animal species and humans. ST88 is known as a typical African clone and has been well described in several Sub-Saharan African countries as a major circulating clone within both hospital and community settings [54]. This clone has also been identified at a lower frequency in Asia and in other parts of the world. In livestock, it has been reported only in Africa, with MRSA ST88 isolated in one pig in Senegal and in healthy sheep in the Ivory Coast [55,56]. More recently, MRSA ST88 was also reported in pigs and pig farm workers in Nigeria [57]. In China, MRSA ST88 has been detected in foodstuff of animal origin and in livestock workers [58,59].

On the other hand, ST6 has been described in MSSA from domestic carnivore nasal swabs (dogs and cats) in Tunisia but also in MSSA nasal swabs from domestic animals (cats, dogs and sheep) and wild animals (monkeys) in many African countries [14,56]. ST6 was also detected in bovine milk in Tunisia [30]. Furthermore, in this context, this clone is believed to be the most dominant sequence type in MSSA associated with food poisoning in China, including milk, meat, etc. [60], suggesting that livestock can be a reservoir of pathogenic bacteria with the ability to cross the host species barrier through their zoonotic capacities, hence becoming a serious public health threat. Movement of camels via trade and contact with farmworkers, veterinarians and tourists should be therefore assessed. 

## 4. Conclusions

This study provides interesting initial genotyping data regarding the nasal and rectal carriage of *S. aureus* in Tunisian dromedaries. Our study highlights that camels’ nostrils and feces should be considered as probable sources of human staphylococcal infections and environmental spread, given their zoonotic capacity. We also underline that the isolates recovered in this study harbored many virulence factors, thus representing a potential threat. In future investigations, it would therefore be interesting to genotype isolates (i) from diseased and healthy animals, including camels from other regions and other domestic animal species in contact with camels, and (ii) from diseased and healthy humans, with a special focus on herdsmen. This could help in understanding the host specificity and assessing the zoonotic potential of *S. aureus.*

## 5. Material and Methods

### 5.1. Study Area and Sampling

The CoPS carriage study was carried out from April 2015 to June 2017 on 159 dromedaries from four Tunisian governorates and five regions (Figure 2): one governorate from Central Tunisia, represented by Kairouan (*n* = 10; semi-arid bioclimatic area; 35°40′41″ N/10°05′46″ E; altitude = 65 m); Sousse, from the coast of Tunisia, including two regions—Hammam Sousse (*n* = 40; semi-arid bioclimatic area, 35°51′39″ N/10°36′11″ E; altitude = 10 m) and Bouficha (*n* = 23; semi-arid bioclimatic area, 36°17′49″ N/10°27′21″ E; altitude = 8 m); and two governorates from Southern Tunisia, represented by Tatatouine (*n* = 32; Saharan bioclimatic area: 32°55′46″ N/10°27′06″ E; altitude = 238 m) and Douz (*n* = 54, 33°275′8″ N/9°01′13″ E; altitude = 62 m). The choice of the sampling area was dictated by the fact that the four selected locations harbor more than 50% of the national cameline population [61].

Samples were randomly collected from apparently healthy animals. From each dromedary, both nasal and rectal mucous membranes were sampled. Commercial sterile cotton-tipped swabs were used for the swabbing, and we proceeded by rubbing the swab against the mucosal surface for approximately 5–10 s at a depth of 5–10 cm into the anterior nares and the rectum. Data concerning herds (zootechnic usage of camels, feeding, herd size) as well as information about dromedaries (gender, age, health status) were checked.

### 5.2. Bacterial Isolation and Identification

We proceeded, using the sampled swabs, to a direct inoculation for enrichment in a brain–heart infusion broth with 6.5% sodium chloride and 10% mannitol concentrations, and the incubation, at 37 °C, lasted 18–24 h. A loopful of each broth inoculum was streaked on the mannitol salt agar (Biokar) selective medium, and the culture plates were incubated at 37 °C for a supplementary duration of 24 h. Positive cultures for *Staphylococcus* spp. were initially identified by conventional methods, including Gram staining and a catalase test. Coagulase-positive and -negative *Staphylococcus* were then respectively determined by whether they had the ability to coagulate rabbit plasma (Biokar).

Among the CoPS, standard biochemical tests were first used to identify *Staphylococcus aureus* with the application of conventional methods [62]. *S. aureus* identification and methicillin resistance/susceptibility were then confirmed through the amplification of the specific genus gene (*sta*), the species-specific thermonuclease gene (*nuc*) and the *mecA* gene by multiplex PCR [24].

### 5.3. Antimicrobial Susceptibility Testing

*S. aureus* isolates were subjected to an in vitro antimicrobial testing method on Mueller-Hinton agar, employing fresh nutrient agar culture and antibiotic discs, according to the performance standards of the Antibiogram Committee of the French Society of Microbiology (CA-SFM 2019). The organisms were left to grow overnight in nutrient agar and adjusted to the 0.5 MacFarland standard, then spread on Mueller-Hinton agar plates and incubated at 37 °C for 18–24 h. A panel of fourteen (14) antimicrobial discs was tested, including penicillin (6 μg), oxacillin (5 μg), cefoxitin (30 μg), ertapenem (10 μg), chloramphenicol (30 μg), tetracycline (30 μg), gentamicin (10 μg), streptomycin (10 μg), vancomycin (5 μg), teicoplanin (30 μg), erythromycin (15 μg), clindamycin (2 μg), sulfamethoxazole (25 μg) and marbofloxacin (5 μg). Moreover, a double-disk diffusion test (D-test) with erythromycin and clindamycin was implemented in all isolates to detect inducible clindamycin resistance. The zones of inhibition around the discs were measured and interpreted as sensitive, intermediate and resistant, according to the interpretation chart of the CA-SFM 2019.

### 5.4. Detection of Staphylococcal Virulence Genes

All isolates were subject to PCR in order to detect one (or more) of the 18 genes coding for staphylococcal enterotoxins (*sea, seb, sec, sed, see, seg, seh, sei, sej, sek, sel, sem, sen, seo, sep, seq, ser* and *seu)* [63]. We also screened the isolates to ascertain the presence of genes encoding for any exfoliative toxins (*siet, eta, etb*), hemolysins (*hla, hlb, hld, hlg* and *hlgv*), the *tst* gene encoding for the TSST, the *luk*S–*luk*F genes encoding for PVL leukocidin, the *luk*E–*luk*D genes encoding the bicomponent leukotoxin LukE–LukD and the *luk*M gene encoding leukocidin M. Adhesin factors, including clumping factors (*clf*A, *clf*B) and fibronectin-binding protein (*fnb*A, *fnb*B); binding proteins, including fibrinogen-binding protein (*fib*), collagen-binding protein (*cbp*), bone sialoprotein-binding protein (*bsp*), laminin-binding protein (*lamin*), elastin-binding protein (*ebp*); and genes encoding capsules (*cap*5, *cap*8) were also tested (Appendix A) [64,65].

### 5.5. Typing of S. aureus Isolates

#### 5.5.1. Spa Typing

*S. aureus* protein A (spa) typing was performed in all *S. aureus* isolates as described by Harmsen et al. (2003) [66]. The polymorphic X region of the *spa* gene was amplified by PCR, and the sequences were analyzed using Ridom Staph-Type software, v.1.5.21 (Ridom GmbH), which automatically detects *spa* repeats and assigns a *spa* type in concordance with the specifications of http://spaserver.ridom.de/ (accessed on 10 May 2021).

#### 5.5.2. Pulsed-Field Gel-Electrophoresis (PFGE)

All *S. aureus* isolates were characterized by pulsed-field gel-electrophoresis (PFGE) employing *Sma*I restriction enzyme digestion, as described by Bouzaiane et al. (2008) [67]. Samples were run on a 1% agarose gel in 0.5 Tris-borate-EDTA (TBE) buffer kept at 14 °C, with the support of a CHEF DR-II PFGE system, using switching times ranging from 5 to 40 s over 20 h at 6 V/cm. The DNA fingerprints generated by PFGE were visually and digitally analyzed according to the Tenover criteria [68] and by GelCompar II 6.6 software, respectively.

#### 5.5.3. Multi-Locus Sequence Typing (MLST)

One representative strain per detected PFGE profile was characterized by multilocus sequence typing (MLST). The allelic profile of each isolate was obtained by sequencinginternal fragments of seven unlinked housekeeping genes (*arcC, aroE, glpF, gmk, pta, tpi* and *yqiL*) [69], in order to determine the sequence type (ST) and clonal complex (CC) assigned by MLST and BURST (Based Upon Related Sequence Types) analyses (www.pubmlst.org, accessed on 15 May 2021).

## Figures and Tables

**Figure 1 animals-11-02754-f001:**
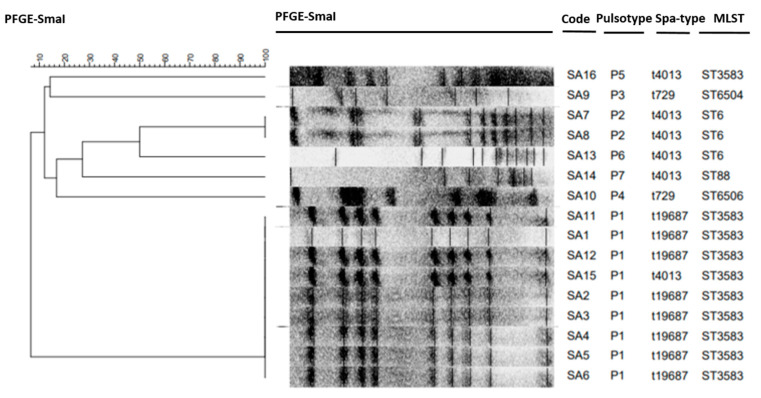
Dendrogram of pulsed-field gel electrophoresis *Sma*I patterns among 16 *Staphylococcus aureus* isolates recovered in this study, generated by GelCompar II software using the unweighted pair group method with arithmetic mean (UPGMA) algorithm and the Dice similarity coefficients.

**Figure 2 animals-11-02754-f002:**
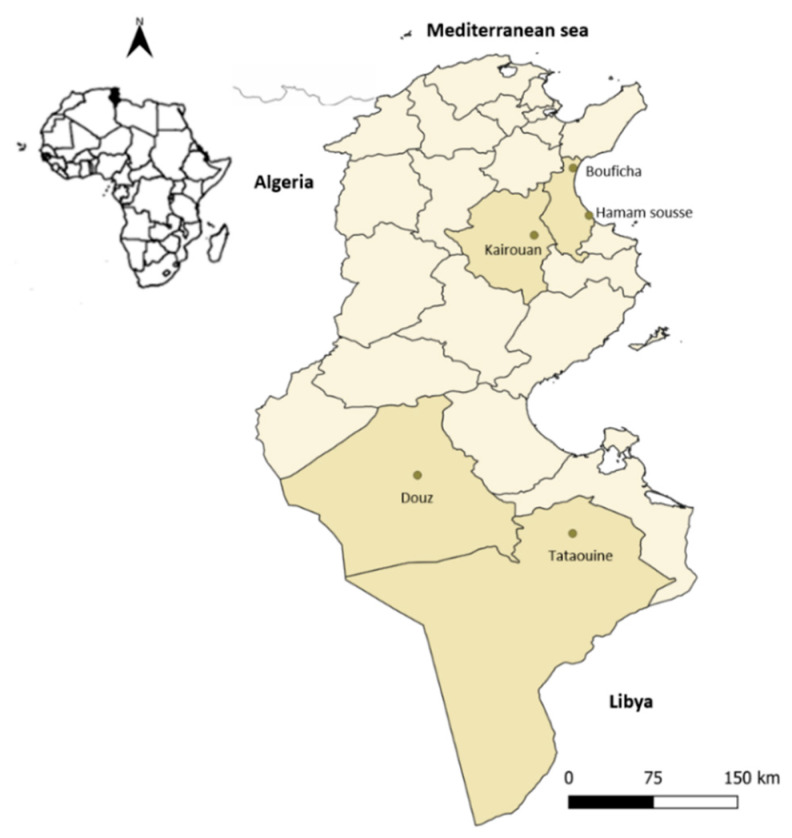
Map of Tunisia showing the visited governorates.

**Table 1 animals-11-02754-t001:** Virulence-associated genes of the 16 MSSA isolated from the healthy camels.

CODE	Hemolysins	Leucocidins	Adhesin Factor	Binding Proteins	Capsular Type
SA1	*hla, hlb, hld*	*lukDE*	*clfA, clfB*	*ebp, lamin*	8
SA2	*hld, hlg2*	*lukDE*	*clfB*	*ebp, lamin*	8
SA3	*hla, hlb, hld, hlg2*	*lukDE*	*clfB*	*fib, ebp, lamin*	8
SA4	*hlb, hld, hlg2*	*lukDE*	*clfA, clfB*	*fib, ebp, lamin*	8
SA5	*hla, hlb, hld, hlg2*	*lukDE*	*clfB*	*fib, lamin*	8
SA6	*hla, hlb, hld, hlg2*	*lukDE*	*clfB*	*fib, ebp, lamin*	8
SA7	*hla, hlb, hld, hlg2*	*lukDE*	*clfB*	*cbp, lamin, ebp*	8
SA8	*hla, hlb, hld, hlg2*	*lukDE*	*clfB, fnbB*	*cbp, lamin, ebp*	8
SA9	*hla, hlb, hld, hlg2*	*lukDE*	*clfA, clfB, fnbB*	*lamin*	8
SA10	*hla, hlb, hld, hlg2*	*lukDE*	*clfB*	*lamin*	8
SA11	*hla, hlb, hld, hlg2*	*lukDE*	*clfA, clfB*	*ebp, lamin*	8
SA12	*hla, hlb, hld, hlg2*	*lukDE*	*clfB*	*ebp, lamin*	8
SA13	*hla, hlb, hld, hlg2*	*lukDE*	*clfB, fnbB*	*ebp, lamin*	8
SA14	*hla, hlb, hld*	*lukDE*	*clfB*	*lamin*	8
SA15	*hla, hlb, hld, hlg2*	*lukDE*	*clfB, fnbB*	*lamin*	8
SA16	*hla, hlb, hld, hlg2*	*lukDE*	*clfA, clfB*	*lamin*	8

**Table 2 animals-11-02754-t002:** Characteristics of the 16 *S. aureus* isolates recovered from 14 healthy camels.

Code	Animal	Sites	Geographic Origin	Spa Type	Pulsotype	ST	Virulence Genes
SA1	1	Nasal	Hamam Sousse	t19687	P1	3583	*clfA, clfB, lukDE, hla, hlb, hld, lamin, ebp, cap8*
SA2	2	Nasal	Hamam Sousse	t19687	P1	3583	*clfB, lukDE, hld, hlg2, lamin, ebp, cap8*
SA3	3	Nasal	Hamam Sousse	t19687	P1	3583	*clfB, lukDE, hla, hlb, hld, hlg2, fib, lamin, ebp, cap8*
SA4	4	Rectal	Hamam Sousse	t19687	P1	3583	*clfA, clfB, lukDE, hlb, hld, hlg2, fib, lamin, ebp, cap8*
SA5	5	Nasal	Hamam Sousse	t19687	P1	3583	*clfB, lukDE, hla, hlb, hld, hlg2, fib, lamin, cap8*
SA6	5	Rectal	Hamam Sousse	t19687	P1	3583	*clfB, lukDE, hla, hlb, hld, hlg2, fib, lamin, ebp, cap8*
SA7	6	Nasal	Hamam Sousse	t4013	P2	6	*clfB, lukDE, hla, hlb, hld, hlg2, cbp, lamin, ebp, cap8*
SA8	7	Nasal	Hamam Sousse	t4013	P2	6	*clfB, lukDE, hla, hlb, hld, hlg2, fnbB, cbp, lamin, ebp, cap8*
SA9	8	Nasal	Hamam Sousse	t729	P3	6504	*clfA, clfB, lukDE, hla, hlb, hld, hlg2, fnbB, lamin, cap8*
SA10	9	Nasal	Hamam Sousse	t729	P4	6506	*clfB, lukDE, hla, hlb, hld, hlg2, lamin, cap8*
SA11	10	Nasal	Hamam Sousse	t19687	P1	3583	*clfA, clfB, lukDE, hla, hlb, hld, hlg2, lamin, ebp, cap8*
SA12	10	Rectal	Hamam Sousse	t19687	P1	36551	*clfB, lukDE, hla, hlb, hld, hlg2, lamin, ebp, cap8*
SA13	11	Rectal	Hamam Sousse	t4013	P6	6	*clfB, lukDE, hla, hlb, hld, hlg2, fnbB, lamin, ebp, cap8*
SA14	12	Nasal	Bouficha	t4013	P7	88	*clfB, lukDE, hla, hlb, hld, lamin, cap8*
SA15	13	Nasal	Bouficha	t4013	P1	3583	*clfB, lukDE, hla, hlb, hld, hlg2, fnbB, lamin, cap8*
SA16	14	Nasal	Bouficha	t4013	P5	3583	*clfA, clfB, lukDE, hla, hlb, hld, hlg2, lamin, cap8*

## Data Availability

The data that support the findings of this study are openly available at www.pubmlst.org (accessed on 30 April 2021).

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
