# Peer review of "First Report of Antimicrobial Susceptibility and Virulence Gene Characterization Associated with *Staphylococcus aureus* Carriage in Healthy Camels from Tunisia"

_animals, 2021, doi:10.3390/ani11092754_

Round 1

Reviewer 1 Report

aureus in camels of tunisia

each commented is preceded by the line number

23 These were isolated from 14 camels…

24 what does “and one isolate/sample was further studied” mean?

99 on the other hand

101 change 

“nor were nine virulence genes, including siet, “

 to 

“nor were the nine virulence genes: siet,….” 

121-125 other studies mentioned involve the current authors.  this should be acknowledged here such as “previously we…”

133 “including Kenya, Iran, Sudan or” change “or” to “and”

135 change “A prevalence of 30.4% and 26.5% was respectively signaled in Sudan and Ethiopia, in camels with pneumonic lesions” to “ In Sudan and Ethiopia camels with pneumonic lesions were also positive for Staphylococcus spp. (30.4% and 26.5%, respectively)

193-196 combine these paragraphs, remove the word “Besides” and “In addition”

203 “camels are sturdy animals and are therefore exceptionally treated” is confusing 

216-218 change “hld was identified in all isolates, it was reported that delta hemolysin is produced by 97% of S. aureus strains (Dinges et al., 2000), which agrees with the studies conducted in Algeria and Saudi Arabia that reported the presence of hld in 100% of camels nasal 218 isolates and meat respectively (Raji et al., 2016; Agabou et al., 2017)”  

to 

“hld was identified in all isolates which is consistent with Dinges et al. (2000) and studies conducted in Algeria and Saudi Arabia that report greater than 97% positivity in camel nasal and meat isolates (Raji et al., 2016; Agabou et al., 2017)”

219 “Besides, hlb was described” to “ hlb has been described”

232-234: Break this sentence up. You have a colon, a semicolon and a few commas all in the same sentence.  Hard to read.

235 Raji et al., 2016; Agabou et al., authors mentioned 2x in this area, no need.

 244 remove “In the other hands”

263 were

258-259 author mentioned 2x

308 remove “yet there are no data available to which our results can be compared.”, you already state that it is the first.  

311-314 “include testing of both diseased AND healthy animals and humans since looking at healthy animals was a sub theme of your work

367-377 Are Hwang, Thompson and Rossato methods being followed (primers, per settings, etc)?  It should be stated as much.

citations: 

citations aren’t appropriate: several cases where a reference is used to support a introductory statements about camels whereas the cited paper is highly specific about Staph aureus (see Graveland, Zhu and Zubair). 

Primary research articles shouldn’t be used to reference a statement in its introduction which can lead to a paper trail.  Instead, support general comments, such as population counts, with review articles.    

Woolhouse doesn’t mention PVL gene

With a hefty amount of genetic analysis there are only 2 figures, one being a map of the area.  The only supplementary material are PCR conditions (which are nice considering M&M lacks lots of detail).  Perhaps a table summarizing all isolates in the Supplementary would be good too.  Conclusion could weave in a statement about how these isolates carry many of the virulence factors associated with disease of this bacterium.

Reviewer 2 Report

The manuscript is well written and demonstrated interesting results. The paper presents the virulence and resistance profile of S. aureus in health camels from Tunisia. The number of S. aureus isolates is low, but the identification of important virulence genes justifies the importance of results.

In my opinion, the manuscript could be improved so much if the virulence and resistance genes would be extended for coagulase-negative strains. So, the paper would be more informative for the field.

In this format, I have some suggestions:

On supplementary material: Correct the name of genes (lower case).

On figure 1, Indicate what is spa type, MLST, and others information on the figure. The dendrogram has to be improved to stay more clear.

I suggest that the authors put the results of genes in a table for a better understanding.

Round 2

Reviewer 2 Report

The authors answered my concerns. I believe that the manuscript could be accept as short communication considering the low number of strains.
